# Cardiac Metabolism and MiRNA Interference

**DOI:** 10.3390/ijms24010050

**Published:** 2022-12-20

**Authors:** Krishnamoorthi Sumaiya, Thiruvelselvan Ponnusamy, Kalimuthusamy Natarajaseenivasan, Santhanam Shanmughapriya

**Affiliations:** 1Medical Microbiology Laboratory, Department of Microbiology, Centre for Excellence in Life Sciences, Bharathidasan University, Tiruchirappalli 620024, Tamil Nadu, India; 2Department of Medicine, Department of Cellular and Molecular Physiology, Heart and Vascular Institute, College of Medicine, Pennsylvania State University, Hershey, PA 17033, USA; 3Department of Neural Sciences, Lewis Katz School of Medicine, Temple University, Philadelphia, PA 19140, USA

**Keywords:** cardiac metabolism, fatty acid metabolism, glucose metabolism, miRNAs, ATP generation, miRNA targeting therapy, cardiometabolic disease

## Abstract

The aberrant increase in cardio-metabolic diseases over the past couple of decades has drawn researchers’ attention to explore and unveil the novel mechanisms implicated in cardiometabolic diseases. Recent evidence disclosed that the derangement of cardiac energy substrate metabolism plays a predominant role in the development and progression of chronic cardiometabolic diseases. Hence, in-depth comprehension of the novel molecular mechanisms behind impaired cardiac metabolism-mediated diseases is crucial to expand treatment strategies. The complex and dynamic pathways of cardiac metabolism are systematically controlled by the novel executor, microRNAs (miRNAs). miRNAs regulate target gene expression by either mRNA degradation or translational repression through base pairing between miRNA and the target transcript, precisely at the 3’ seed sequence and conserved heptametrical sequence in the 5’ end, respectively. Multiple miRNAs are involved throughout every cardiac energy substrate metabolism and play a differential role based on the variety of target transcripts. Novel theoretical strategies have even entered the clinical phase for treating cardiometabolic diseases, but experimental evidence remains inadequate. In this review, we identify the potent miRNAs, their direct target transcripts, and discuss the remodeling of cardiac metabolism to cast light on further clinical studies and further the expansion of novel therapeutic strategies. This review is categorized into four sections which encompass (i) a review of the fundamental mechanism of cardiac metabolism, (ii) a divulgence of the regulatory role of specific miRNAs on cardiac metabolic pathways, (iii) an understanding of the association between miRNA and impaired cardiac metabolism, and (iv) summary of available miRNA targeting therapeutic approaches.

## 1. Introduction

The heart is a metabolic omnivore capable of processing multiple energy-providing substrates, including fatty acids (FA), glucose, lactate, ketone bodies, and amino acids simultaneously to maintain the uninterrupted production of chemical energy, adenosine triphosphate (ATP), to support its continuous contractile process [1,2]. The heart has an extremely high energy requirement, and six kilograms of ATP are utilized and produced by the human heart daily. Cardiac metabolism is an integral part of cardiac function as the energy utilizer and producer of the heart. Cardiac metabolism is coordinately regulated by energy demand, the availability of metabolic intermediates, and the transcriptional, translational, and post-translational regulation of enzymatic functions [3]. As cardiac activities are highly dependent on ATP generation, perturbations in the cardiac metabolic process lead to catastrophic consequences on cardiac function. So, the derangement of cardiac metabolism plays a crucial role in several cardiac diseases. Therefore, manipulations in cardiac energy metabolism in heart failure patients have potential therapeutic relevance. All things considered, attaining massive knowledge of the exact regulatory system of cardiac metabolism will help to understand several heart diseases and develop novel therapeutic paths. In this review, we discuss reports investigating the cardiac metabolic pathways and their regulation, concentrating mainly on the understanding of miRNA-regulated cardiac metabolism and miRNA-dependent novel therapeutic avenues for the treatment of heart failure patients.

A total of 95% of the ATP requirements of myocardia were accomplished by oxidative phosphorylation of mitochondria and the residual 5% by glycolysis. Approximately 70% to 90% of mitochondrial ATP was generated from fatty acid oxidation, and the remaining ATP resulted from pyruvate oxidation (derived from glucose and lactate), a trivial amount of ketone bodies, and some amino acids [4]. Because of the metabolic flexibility of the heart, the myocytes can readily shift between fuel sources, and alterations in the predilection of energy states are strongly dependent on the availability of energy substrates, the workload of the heart, and hormonal and neuronal activities [5,6,7]. In a diseased state, a shift of substrate selectivity occurs [8]. During pathologies, an impaired utilization and oxidation of fatty acid and enhanced fuel reliance on glucose occurs, leading to diminished cardiac oxidative metabolism and reduced production of ATP [9]. Oxidation of ketone bodies and amino acids in a failing heart is heightened, and the oxidation of ketone bodies can suppress the oxidation and uptake of glucose and fatty acids [5,10].

A growing body of evidence indicates that cardiac metabolism is systematically regulated by several significant executors such as peroxisome proliferator-activated receptors (PPARs) including PPARα and PPARγ [11], adenosine monophosphate-activated kinase (AMPK) [12], noncoding RNAs (ncRNAs) including microRNA (miRNA) [13] and long noncoding RNA (lncRNA) [14], stromal interaction molecule (STIM) [15], Ca^2+^/calmodulin-dependent protein kinase II (CAMKII) [16], Protein kinase C [17], pyridine nucleotides including NAD^+^ and NADP^+^ [18]. PPAR is a class of ligand-activated nuclear receptors and the primary transcription factor that regulates FA oxidation (FAO). PPARα, an isoform of nuclear hormone receptor, acts as a potent regulator of lipid and glucose homeostasis by transcriptionally regulating the enzymes associated with fatty acid uptakes such as CD36, Diacylglycerol acyltransferases (DGAT), Carnitine palmitoyltransferase 1 (CPT-1), and acyl-CoA dehydrogenase and it also controls the mitochondrial FA oxidation [19]. Another essential regulator, ncRNA, plays a crucial role in cardiac function by regulating gene expression. The regulation of most of the cellular functions is correlated with ncRNAs. Specifically, miRNAs have emerged as a prominent player in cellular functions and pathophysiology by regulating about half of all eukaryotic mRNAs at the post-transcription level through binding and targeting mRNAs for degradation [20]. miRNAs are small, tissue-specific RNA comprising 19–25 base pairs. Over 2000 human miRNAs have been cataloged so far [21] miRNAs have accumulated considerable attraction for their potential to organize modifications to the transcriptome and eventually proteome during heart diseases. Interestingly, miRNA has been reported to regulate energy homeostasis and metabolism in the cardiovascular system [13]. The functionally efficient miRNAs, including miR-199a, miR181c, miR-214, miR-378, miR-378*, miR-24, miR-126, miR-21, miR-21, and miR-29 target the PPAR family receptor and PPAR γ coactivator- 1 (PGC1) to regulate the switch in substrate utilization from fatty acids to glucose. These miRNAs can control the differential expression of proteins involved in substrate transport, β-oxidation, glycolysis, metabolism of ketones, Krebs cycle, and electron transport chain [22].

Abnormalities in the cardiac metabolic pathways are a potential risk factor for several cardiometabolic diseases. As fatty acids and glucose are the predominant energy-providing substrates for cardiac function, defects in the metabolism of these substrates prevalently play a vital role in most cardiometabolic diseases. The potential inhibitory role of miRNAs in cardiac metabolism makes them a key player in cardiometabolic diseases. Recent works in the literature have discussed the role of circulating miRNAs as biomarkers of cardiovascular diseases [23,24] and the prominent therapeutic targets of several pathophysiological conditions [14,25,26].

Previous review analyses on the contribution of miRNA in cardiac metabolism aimed to review the knowledge of miRNA interference [27,28,29]. Likewise, our study intends to exhibit the recent advances in knowledge on the interplay between miRNAs and cardiac metabolism and abnormalities. Most of the growing evidence has shown the regulatory role of individual miRNAs in cardiac metabolism and their potential to serve as biomarkers and therapeutic targets for cardiac diseases. In addition, the existing evidence has focused on the role of miRNAs in only the potent energy substrate metabolism [30,31,32]. Further to this, this review collectively provides a panel of multiple relevant miRNAs based on the available evidence that can be employed in future research on the combinatorial effects of multiple miRNAs in the field of diagnosis and prognosis and therapeutics to attain higher accuracy and efficacy, respectively.

## 2. Methods

A comprehensive literature search was conducted through electronic databases such as PubMed, Scopus, EMBASE, and ScienceDirect to identify the articles that elucidated the miRNA interference in cardiac metabolism, diagnosis, and therapeutics of cardiometabolic diseases. The databases were searched up to and including October 2022 by using key terms including “cardiac metabolism,” “metabolic pathways of cardiac energy providing substrates,” “regulation of cardiac metabolism by miRNA,” “miRNA associated cardiometabolic diseases,” and “miRNA targeted therapeutic strategies”. The identified articles were screened for relevance using the title and abstract, and the full text of relevant articles was retrieved. The cited articles of most relevant studies were also searched to obtain more reports. Journals without a science citation index or impact factor were excluded as were conference abstracts and non-English language articles. Searching the screened results assembled 172 articles that were used to extract scientific data to write the review systematically.

## 3. Cardiac Metabolism

Before the regulatory role of miRNA in cardiac metabolism can be fully appreciated, it is essential to understand the mechanism of cardiac metabolism. The cardiac metabolism of the human heart comprises the transfer of energy, regulation of every substrate metabolism, and the control of enzymatic activity at transcriptional, translational, and post-translational levels [33,34]. The molecular pathways of the cardiac fuel metabolism and distribution system are more intricate and active than electrical power grids. Due to the low potential of the heart to store energy substrates within cells, it obtains fuel continuously from the blood. Each energy-providing substrate of the heart has a difference in ATP yield, calculated by the molarity or oxygen consumption per ATP production. Fatty acids provide a higher yield of ATP per molecule of metabolized energy substrate than glucose, whereas glucose provides 40% more ATP production per molecule of oxygen consumed than fatty acids that require a 15% higher amount of oxygen than glucose [35,36,37].

Energy substrates are moved into the cytosol by the extracellular membrane transversely and are then metabolized in different steps. The metabolism of energy-providing substrates by the heart is assembled into four stages. The first stage comprises substrate delivery and uptake by the cells. The second stage consists of cytosolic and mitochondrial pathways that form a central metabolic intermediate, acetyl-coenzyme A (acetyl-CoA). The third stage involves the oxidation of acetyl CoA in the Krebs cycle to produce reducing equivalents, namely, nicotinamide adenine dinucleotide (NADH_2_) and flavin adenine dinucleotide (FADH). The fourth stage consists of the production of ATP by the electron transport chain [3,38]. In this part of the review, we discuss the metabolic and oxidative processes of cardiac energy-providing substrates for the generation of ATP.

### 3.1. Fatty Acid

The heart uses many fatty acids (FAs) as energy substrates for ATP generation. The tight coupling of fatty acid uptake with its oxidation leads to the failure of excess lipid accumulation in cardiomyocytes [39]. Free fatty acids (FFAs) bound with albumin or triacylglycerol of chylomicron-derived fatty acids are the primary source of fatty acids supplied to the heart. The entry of FFAs into the cardiomyocytes is promoted either by passive diffusion or by transport proteins such as fatty acid translocase (CD36/SR-B2), a scavenger receptor class B protein, and fatty acid-binding protein (FABP) [40]. Following uptake by cardiomyocytes, FFAs are bound by a heart-specific FABP and are activated by esterification to form fatty acyl-CoA in the cytosol by using fatty acyl- CoA synthetase (FACS). The fatty acid moiety of fatty acyl CoA is then converted to long-chain acylcarnitine by the outer mitochondrial membrane carnitine palmitoyl transferase 1 (CPT-I). The resulting acylcarnitine is transported into the mitochondrial matrix, where its long-chain fatty acids are again converted to long-chain acyl-CoA by CPT-II. Within mitochondria, the long-chain acyl CoA enters β-oxidation and generates acetyl CoA and reducing equivalents. The primary products of fatty acid oxidation (FAO) are FADH_2_, NADH, and acetyl CoA, which give rise to more FADH_2_ and NADH production through the Krebs cycle [41,42,43]. In heart failure, the switching of energy substrate metabolism from FAO to high-extent metabolism of glucose was evidenced [44].

### 3.2. Glucose

The heart utilizes either exogenous glucose or a derivative of stored glycogen. Glucose is considered an essential energy substrate for the heart to produce chemical energy from glycolysis in cytoplasm and the oxidation of glycolytic-derived pyruvate in mitochondria [45]. As glucose undergoes glycolysis and facilitates the anaerobic generation of ATP, glucose is considered the most efficient energy substrate of the heart. Glucose uptake by cardiomyocytes occurs along a steep concentration gradient by isoforms of specific transmembrane glucose transporters, namely, GLUT1 and GLUT4 [46,47]. After cardiomyocytes take up glucose, the cytosolic glucose is rapidly phosphorylated to glucose 6-phosphate (G6P) by hexokinase II, the cardiac isoform of hexokinase. G6P, a metabolic intermediate, is at the branch point of many metabolic pathways: glycolysis converting the glucose into pyruvate, the glycogen synthase reaction to produce glycogen, the pentose phosphate pathway (PPP), which oxidizes G6P into ribose and nicotinamide adenine dinucleotide phosphate (NADPH), and the hexosamine biosynthetic pathway (HBP). Glycolysis represents the primary pathway in glucose use in the heart, which produces pyruvate, NADH, and a minimum amount of ATP. The glycolysis-derived pyruvate can either be reduced into lactate or be entered into mitochondria by the mitochondrial pyruvate carrier (MPC). Inside the mitochondrial matrix, pyruvate is decarboxylated into acetyl-CoA by multienzyme complex pyruvate dehydrogenase (PDH) enzyme, which resides in the inner mitochondrial membrane, and subsequently, the resultant acetyl-CoA is then fed into the Krebs cycle. In another way, pyruvate is also carboxylated into oxaloacetate, which plays a crucial role in anaplerosis. PDH is activated and inactivated by specific enzymes, PDH phosphatases and PDH kinases, respectively. Furthermore, the increased ratio of acetyl-CoA/CoA and NADH/NAD^+^ rapidly activates the PDH [35,45,48,49]. In a state of higher energy metabolic demand or heart failure, glucose may act as a prominent energy-providing substrate for ATP production [45]. During ischemia, decreased aerobic energy production enhances glucose uptake, glycolysis, and the catabolism of glycogen, causing glycolysis to be an essential source of energy production [34].

### 3.3. Lactate

Cardiac lactate metabolism is an effective process that aids in adapting to changes in energy requirements. It acts as a multifunctional molecule such as an energy substrate, metabolite, signaling molecule, and prognostic factor. During certain pathophysiological conditions of the heart, lactate metabolism switches the energy substrate requirements. Lactate enters the cardiomyocytes through monocarboxylic anion transporter 4 (MCT4) and is converted into pyruvate using lactate dehydrogenase (LDH). The resultant pyruvate undergoes a similar fate as the glycolytic-derived pyruvate described in Section 3.2 [50,51].

### 3.4. Ketone Bodies

d-β-hydroxybutyrate (βOHB) and acetoacetate (AcAc) are important ketone bodies as energetic substrates, which can be metabolized by the heart, especially in carbohydrate-deficient conditions as glucose-sparing energy-providing fuels [52]. Ketone bodies are produced in the liver by using acetyl-CoA from FAO. Acetyl-CoA is converted into AcAc through sequential steps in a metabolic process called ketogenesis and by using acetoacetyl-CoA thiolase (ACAT), hydroxymethylglutaryl-CoA synthase (HMGCS), and HMG-CoA lyase (HMGCL). AcAc is then transformed into βOHB by β-hydroxybutyrate dehydrogenase (BDH) [53]. βOHB and AcAc are readily transported into cardiomyocytes by monocarboxylate transporter (MCT). Once in the mitochondria, βOHB is oxidized to AcAc by mitochondrial BDH, and the AcAc is then activated by succinyl-CoA:3-ketoacid-CoA transferase (SCOT) to form the acetoacetyl-CoA. Further, the acetoacetyl-CoA is cleaved into two molecules of acetyl-CoA, which are then fed into the Krebs cycle to be turned into ATP through oxidative phosphorylation [53,54,55,56]. In patients with heart failure with reduced ejection fraction (HFrEF), the myocardial uptake of ketone bodies was increased by upregulating the expression of SLC16A1 (a monocarboxylate transporter), with a concomitant increase in the oxidation of ketone bodies [57,58].

### 3.5. Amino Acids

Other than being well-known as the building blocks of proteins, amino acids are also an essential part of the energy metabolism of the heart. Glutamate and aspartate are vital amino acids that contribute to the transportation of reducing equivalents and hydrogen ions into mitochondria for the oxidation of NADH by ETC [59]. Oxidation of branched-chain amino acids (BCAA), including leucine, isoleucine, and valine, is a vital source of cardiac amino acid oxidation [60]. After the uptake of BCAA by cardiomyocytes, branched-chain aminotransferase (BCAT) avidly transaminates the BCAA into branched-chain α-keto acids (BCKA). Then, the mitochondrial branched-chain α-keto acid dehydrogenase (BCKDH) catalyzes the oxidative decarboxylation of BCKA to acetyl-CoA and succinyl-CoA for tricarboxylic acid (TCA) cycle and anaplerosis, respectively. Although the oxidation of BCAA is a trivial source of cardiac energy production, it can modify the cardiac signaling pathways such as insulin and mammalian target of rapamycin (mTOR) signaling [61,62].

## 4. Regulation of Cardiac Metabolism

Cardiac metabolic pathways are regulated by a molecular network of multiple metabolic enzymes that contribute to the utilization of energy-providing substrate and oxidative phosphorylation for ATP production. The selection and use of cardiac substrates are tightly regulated between and linked with two significant substrates, FA and glucose. The utilization of one substrate at a time can directly inhibit the other. Based on the nature of the cardiac disease, the cardiac metabolism may have relied on either oxygen-efficient glucose or fuel-efficient FAs. Insulin, AMPK, glucagon-like peptide (GLP-1), pyruvate, and catecholamines are the other metabolic regulators that optimize cardiac metabolism. Differential expression of and post-translational modifications in metabolic enzymes may firmly regulate the cardiac metabolic processes. PPAR, estrogen-related receptor (ERR)α, PGC-1, and AMPK are efficient mediators under physiological or pathological conditions [1,63,64]. Recent studies suggest that several miRNAs regulate cardiac metabolism by upregulating or downregulating the expression of principal requisites of different metabolic pathways [22]. Therefore, we summarized miRNAs’ mechanistic role and their specific targets in cardiac metabolism (Figure 1).

### 4.1. Regulatory Role of miRNA on Cardiac Metabolism

Over the last two decades, investigations have identified miRNAs’ critical roles in cardiac metabolism and heart diseases. miRNAs blunt target gene expression by base pairing the seed sequence of miRNA, a conserved heptametrical sequence in the 5’ end of the mRNA target gene, especially with the respective recognition sequences which reside prominently in the 3’ untranslated region (UTR) [26]. miRNAs regulate several metabolic pathways by altering the gene expression of eminent metabolic enzymes, metabolites, or transcription factors through mRNA degradation and translational repression. Specific miRNAs are predominantly expressed in the heart, including miR1, miR133a, and miR-208, and maintain cardiac function [65]. In this part of the review, we summarize the present knowledge of the regulatory role of miRNA in cardiac metabolism, including glucose, fatty acids, and amino acids, based on its importance and research attention.

#### 4.1.1. MiRNA-Regulated Glucose Metabolism

Overwhelming evidence suggests that miRNAs modify cardiac glucose metabolism by regulating glucose transport and metabolism and insulin signaling pathways.

##### Glucose Uptake: MiRNA Interference

In the heart, miRNAs have fine-tuned the expression of the glucose transporter, GLUT4. GLUT4 is vital in enhancing glucose uptake upon insulin induction and governing cardiac ATP supply [66]. The role of miRNAs in GLUT4 regulation depends on the direct post-transcriptional regulatory effects on the GLUT4 encoding gene, solute carrier family 2 member 4 (Slc2a4) expression; direct regulatory effects on the targets which maintain Slc2a4 expression, and positive correlation between the miRNA and GLUT4 expression [67]. In silico evidence shows that the 3’ UTR of GLUT4 messenger RNA holds the miRNA binding sites specific to miR-17, miR-20b, miR-93, miR-105, miR-106b, miR-150, and miR-291a [68]. The overexpression of miR-223 significantly increases the PI3K independent glucose uptake by post-transcriptional upregulation of GLUT4 expression at the plasma membrane without altering the expression of GLUT1 [69]. miR-133 and miR-223-3p decreased the protein levels of Kruppel-like zinc finger transcription factor 15 (KLF15) by directly repressing gene expression and subsequently suppressing the expression of downstream target GLUT4. Therefore, silencing miR133a and miR133b may increase the myocardial glucose uptake in heart failure patients [70,71]. miR-200a-5p inhibits the stress-associated selenoproteins (Sel), including selenoprotein 15 (Sep15), selenoprotein t (Selt), and selenoprotein p1 (Sepp1), impairs the Sel-mediated glucose metabolism, increases the glucose uptake in cardiomyocytes and leads to cardiac hypertrophy [72]. During cardiac hypertrophy, markedly decreased levels of miR-1 and miR-133a mitigate the downregulation of GLUT4 expression by increasing the activity of insulin-like growth factor (IGF-1) and Akt signaling, which promotes the phosphorylation of GLUT-4 translocation mediator Akt substrate of 160 kD (AS160) to promote the transport of glucose by GLUT4 [65,73]. miR-150 directly binds to the 3’ UTR of the Slc2a4 gene, represses GLUT4 expression, impairs glucose uptake and consumption in insulin-resistant (IR) cardiomyocytes, and shortens the cardiac energy supply. Consistently, the knockdown of miR-150 and small molecule (AMO-150) mediated inhibition significantly heightens the glucose uptake in insulin resistance cardiomyocytes [68]. The bioinformatics and proteomics analysis of recent evidence found that miR-29c expression in cardiomyocytes is inversely correlated with GLUT4 expression [74]. The expression of miR-17 was upregulated in diabetic cardiac myopathy [75]. We, therefore, speculated that GLUT4, KLF15, selenoproteins, and IGF-1 might be the key targets of specific miRNAs, including miR-223, miR-133, miR-223-3p, miR-200a-5p, miR-1, miR-133a, miR-133b, miR-150, and may regulate the transport and uptake of glucose in cardiomyocytes.

##### Glucose Metabolism and Glycogen Synthesis: MiRNA Interference

After cardiomyocytes take up glucose, they undergo glycolysis to produce ATP. miRNAs play an essential role in the glycolysis of normal cardiomyocytes. The resulting pyruvate is transported into mitochondria to form an essential metabolic intermediate acetyl-CoA and is then fed into the TCA cycle. miR-125b is known to directly target HK2 and reduce the metabolic rate of glucose under hypoxia. In this regard, the inhibition of miR-125b protects cardiomyocytes from hypoxia-stimulated injury by restarting the glucose metabolism [76]. miR-34a and miR-125b significantly decreased intracellular glucose metabolism and lactate production by directly interacting with HK2 and lactate dehydrogenase-A (LDHA) [77]. miR34a also targets the pro-survival protein sirtuin 1 (SIRT1) and induces cardiomyocytes’ apoptosis in the diabetic heart [78]. The tumor suppressor miRNAs, *let-7*, plays a vital role in regulating glucose metabolism. The RNA binding proteins *Lin28a*/*b* inhibit the synthesis of *let-7*; however, the tissue-specific loss of Lin28a or overexpression of *let-7* suppresses IGF1R, insulin receptor (INSR), and insulin receptor substrate 2 (IRS2) of the insulin- PI3K-mTOR pathway which leads to insulin resistance and impaired glucose tolerance. Thus, the *Lin28*/*let-*7 pathway plays a specific and integral role in modulating glucose metabolism [26,79]. Oxygen-glucose deprivation mediated miR-21 plays an essential role in glucose metabolism and cellular glycolysis by heightening the phosphofructokinase activity and cellular glycolysis and acts as a cardioprotective downstream target of the light-elicited circadian rhythm protein Period 2 (Per2). miR-21 expression is required to protect cardiomyocytes against myocardial infarction [80,81]. miR-143 modifies insulin-mediated Akt activation and glucose homeostasis by downregulating the oxysterol-binding-protein-related protein 8 (ORP 8); therefore, the miR-143-ORP8 pathway acts as a critical therapeutic target for obesity-induced diabetes [82]. Targeting hepatocyte nuclear factor-1 beta (HNF-1β) by miR-802 causes impaired glucose tolerance and insulin signaling and promotes gluconeogenesis [83]. miR-802-5p targets heat shock protein 60 (Hsp60), induces insulin resistance, and impairs cardiac function [84]. The abnormalities in cardiac metabolism lead to type 2 diabetes (T2D), obesity, and heart failure. One of the significant metabolic regulatory systems in the heart is MED13, a subunit of the mediator complex, which is counter-regulated by heart-associated miR-208a. The MED13/miR-208a system enhances resistance to aberrant cardiac metabolism-mediated obesity and promotes glucose tolerance and systemic insulin sensitivity [85]. In obese mice, miR-103 and miR-107 get upregulated and induce the impairment of glucose homeostasis and uptake by targeting the critical regulator of the insulin receptor, caveolin-1 [86]. Recent studies revealed that miR-103 and miR-107 regulate systemic glucose metabolism and insulin sensitivity by altering the mitochondrial volume and protein levels of OXPHOS complexes [87]. Nuclear factor erythroid-2–related factor 2 (NRF2) regulated miR-1, and miR-206 downregulated the expression of metabolic genes, including glucose-6-phosphate dehydrogenase (G6PD), 6-phophogluconate dehydrogenase (6PGD), and transketolase (TKT) and ribose synthesis by regulating the pentose phosphate pathway [88]. The ectopic expression of miR-195 also targets the INSR to impair glycogen synthesis [89]. miR-199a targets glycogen synthase kinase 3*β* (GSK3*β*)/mTOR complex signaling and activates the basal cardiomyocyte autophagy to impair cellular homeostasis and glycogen synthesis [90]. In addition, miR-26, miR-378, miR-29c-3p, miR-144-3p, miR-195a-3p, and miR-126 were also reported to target GSK3β to synthesize glycogen [22].

##### MiRNA Regulated TCA Cycle

Pyruvate is the end product of glycolysis transported into mitochondria for the generation of ATP. The Krebs cycle is an integral route for cellular oxidative phosphorylation, which relies on ETC to replenish cells’ redox balance and energy requirements. miR-195 directly targets and downregulates the expression of SIRT3 and, thus, the enzymatic activity of PDH complex and ATP synthase by enhanced acetylation [91]. miR-210 targets the glycerol-3-phosphate dehydrogenase to exhibit cardiac dysfunction by inhibiting mitochondrial oxygen consumption, increasing glycolytic activity, and decreasing mitochondrial reactive oxygen species (ROS) flux [92]. miR-161a and miR-183 target the enzymes of the TCA cycle, isocitrate dehydrogenase 1 and 2 [93]. miR152, miR-494, and miR-19a control the gene expression of the citrate synthase gene by targeting the hypoxia-inducible factor 1-alpha (HIF1α) [94]. The upregulation of miR-499 and miR-761 and the downregulation of miR-140 inhibit the mitochondrial fission and apoptosis of cardiomyocytes by targeting the mitochondrial fission factor (MFF) [95]. miR-30 can inhibit mitochondrial fission via the repression of p53 expression and its downstream target dynamin-related protein-1 (DRP-1) [96].

#### 4.1.2. MiRNA-Regulated Fatty Acid Metabolism

Fatty acid uptake by cardiomyocytes, β-oxidation in mitochondria, the TCA cycle, and the process of ATP production were tightly governed by both cytosolic and mitochondrial enzymes under miRNAs’ control. miR-122 is the first miRNA associated with the regulation of fatty acid metabolism.

##### Fatty Acid Uptake by Cardiomyocytes: MiRNA Interference

Fatty acid uptake by cardiomyocytes is achieved by passive diffusion or protein carrier, namely, fatty acid translocase (FAT/CD36/SR-B2), which facilitates fatty acid transport across the plasma membrane. Several miRNAs target the mRNA of CD36 and post-transcriptionally regulate the target gene expression. The 3’ UTR of CD36 mRNA has binding sites for several miRNAs, and the expression of CD36 is linked to the upregulation of miR-130a, miR-134, mi-141, miR-16, miR-199a, miR-363, miR-15b, miR-22, and 185 and downregulation of miR-152, and miR-342-3p [97,98]. The muscle-specific miRNA-1 binds to its direct non-canonical target, heart-type FA binding protein- 3 (FABP3), a small cytoplasmic protein associated with fatty acid uptake in cardiomyocytes. Immunoprecipitation of biotinylated miR-1 oligo with FABP3 mRNA and Western blotting analysis confirmed the binding interaction in vivo. The myocardial miR-1 expression is inversely correlated with both intracellular and secreted levels of FABP3 by the IGF-1/miR-1/FABP3 axis [99]. miR-223-3p targets the KLF15, which promotes glucose uptake and lipogenesis, notably, the coordinative activity of KLF15 and PPARα regulates the gene expression and oxidation of fatty acids in cardiomyocytes [71]. Lower expression of miR-200b-3p allows the overexpression of SR-B2 in diabetic cardiomyopathy resulting in increased fatty acid uptake, lipotoxicity, and cardiomyocyte apoptosis by activation of the PPAR-γ signaling pathway [100]. However, the targeting interaction between miR-200b-3p and BCL2L11 has recently shown a proactive role against myocardial infarction-induced apoptosis [101]. miR-320 increased fatty acid uptake by interacting with the critical target gene CD36 and causing cardiac lipotoxicity in the diabetic heart. Surprisingly, the miR-320 of the diabetic cardiomyocytes translocated into the nucleus, which assists in upregulating the transcription of CD36; therefore, miR320 acts as a small transcription-activating RNA in the nucleus. It accelerates diabetes mellitus-mediated myocardial stress and dysfunction by enhancing insulin resistance, CD36, and VLDLR (very low-density lipoprotein receptor) expression [90].

##### Fatty Acid Metabolism and Storage: MiRNA Interference

β-oxidation of fatty acids in the mitochondria of cardiomyocytes is firmly regulated in every step of the process by miRNAs. miR-33a and miR-33b regulate the mitochondrial FAO by controlling the respective genes, including carnitine O-octanoyl transferase (CROT), carnitine palmitoyltransferase 1A (CPT1A), and hydroxyacyl-coenzyme A-dehydrogenase (HADHB) and an essential lipid metabolic regulator AMP-activated protein kinase (AMPKα1). In addition, miR 33a/b and its host gene, sterol regulatory element-binding protein family of transcription factors (SREBPs), coordinately govern the gene expression associated with lipid biosynthesis and uptake [102]. miR-27a targets the regulators of lipid metabolism such as SREBP1, SREBP2, PPARα, and PPARγ [103]. miR-197 and miR-146b are upregulated during right ventricular (RV) heart failure, suppressing FAO genes such as *Cpt1b* and *Fabp4* in primary cardiomyocytes [104]. miR-370 and miR-122 target and activate the transcription factor, SREBP-1c, and the fatty acid metabolic enzymes such as DGAT2, fatty acid synthase (FAS), and acetyl-CoA carboxylase 1 (ACC-1). miR-122 reduces the plasma levels of triglyceride and cholesterol by regulating the target genes, including 3-hydroxy-3-methylglutaryl-coenzyme A reductase (HMGCR), HMGCS1, and 7-dehydrocholesterol reductase (DHCR7) [91]. miR-370 binds the 3’ UTR region of CPT1α, represses its activity, and thus the rate of β-oxidation [105]. miR-132 and miR-212 interact with their direct target Carnitine-acylcarnitine translocase (CACT), which transports the long-chain acylcarnitine into mitochondria [106]. miR-30c modulates its target PGC-1β and protects the diabetic cardiomyocytes from apoptosis and cardiac dysfunction by decreasing reactive oxygen species (ROS) production, glucose utilization, and lipid accumulation [107,108]. PGC-1α is a specific target of miR-696, which downregulates mitochondrial synthesis and the rate of FAO and upregulates aerobic metabolism [109]. Carnitine O-acetyltransferase (CRAT) and mediator complex subunit 13 (MED13) are the targets of miR-378 that counterbalance the metabolic actions of the host gene, PGC-1β. miR21 targets FABP7, PPARα and insulin-like growth factor-binding protein 3 (IGFBP3) and regulates lipid metabolism [110]. miR-758 regulates the ATP binding cassette subfamily A member 1 (ABCA1) to maintain cellular cholesterol efflux [111]. Cardiac-specific miR-125b, the first miRNA to be identified, upregulates the gene products involved in fatty acid metabolism [112]. miR-483-3p targets the translational repression of growth/differentiation factor-3 and reduces the lipid storage in adipose tissue leading to lipotoxicity and elevating susceptibility to metabolic diseases [113].

#### 4.1.3. MiRNA Regulated ETC and ATP Generation

The reducing equivalents (FADH and NADH_2_) obtained from the metabolism of energy-providing substrates enter into the ETC for oxidative phosphorylation to generate a vast amount of ATP. The heart has an increased number of mitochondria to produce the chemical energy needed for the continuous contractile process. miR-15b, miR-16, miR-195, and miR-424 decrease the cellular levels of ATP. miR-15b targets the ADP-ribosylation factor-like 2 (Arl2), which resides in adenine nucleotide transporter 1, an ADP/ATP exchanger of mitochondria. miR-15b promotes mitochondrial degeneration and decreases cellular ATP generation by downregulating the mRNA and protein levels of Arl2 [114]. miR-210 downregulates the mitochondrial function and, in turn, enhances glycolysis by inhibiting the potential targets, namely, iron-sulfur cluster scaffold homolog (ISCU) and cytochrome c oxidase assembly protein (COX-10), which are the critical factors of ETC [115]. During type I diabetes, miR-141 negatively regulates the solute carrier family 25 member 3 (Slc25a3), which is important for the transportation of an inorganic phosphate into the mitochondrial matrix for ATP synthesis [116]. Following diabetic insult, the mitochondrial miR-378 expression in interfibrillar mitochondria (IFM) decreased the ATP synthase F_0_ component ATP6 leading to negative regulation of energy metabolism [117]. miR-146a downregulates the mitochondrial ATP production in cardiomyocytes by targeting the dihydrolipoamide succinyltransferase (DLST). DLST is the mitochondrial protein with three eminent recognition sequences for the miR-146a seed sequence in 3’UTR [118]. miR-181c is encoded in the nucleus and then translocated into the mitochondria of cardiomyocytes by natural movement across the mitochondrial membrane via a chaperone, Argonaut 2 (AGO2); it is diffused by polynucleotide phosphorylase (PNPase) and regulates the mitochondrial gene expression. miR-181c has been linked to cardiomyocyte mitochondrial dysfunction by targeting mitochondrial COX-1 (mt-COX-1) mRNA. The inhibition of mt-COX-1 inversely regulates mt-COX-2 expression leading to the derangement of mitochondrial respiratory complex IV; subsequently, it causes enhanced mitochondrial respiration and ROS followed by mitochondrial dysfunction [119,120].

#### 4.1.4. Amino Acid Metabolism Regulated by MiRNA

miR-23b*, miR-29a, and miR-277 are the key miRNAs that participate in amino acid metabolism primarily by regulating acyltransferase and α-ketoacid dehydrogenase [121]. miR-29b regulates the BCAA catabolism by targeting the component of the branched-chain α-ketoacid dehydrogenase complex [122]. miR-122 targets the high-affinity cationic amino acid transporter (CAT-1) and controls the metabolism of the amino acid [123]. miR-200c targets and dysregulates glutamine metabolism by inhibiting glutaminase upon ischemia/reperfusion (IR) injury. Glutamine is a potent amino acid that controls several essential cellular processes, including redox homeostasis and energy formation. Among the miR-200c, miR-21, miR-126, and miR-18 were upregulated and downregulated in the IR heart, respectively. miR-200c plays a vital role in IR injury [124].

#### 4.1.5. Polyamine Metabolism Regulated by MiRNA

Polyamines are insulin-like organic compounds. They modify glucose and fatty acid metabolism; thus, polyamines are principal metabolites of interest. Like insulin, polyamines inhibit the diabetes-mediated upregulation of glucose and ketone bodies. Polyamines such as spermine play a key role in maintaining mitochondrial energetics in an increased glucose environment. The polyamine biosynthetic pathway resides upstream of the TCA cycle and is associated with several metabolic pathways. Polyamine uptake by cardiomyocytes is carried out by SLC3A2, SLC18B, and ATP13A3 [125,126]. miR-34a, miR-34b, and miR-485-3p are upregulated in diabetes mellitus- cardiomyopathy (DMCM) and target the upstream metabolites of polyamine metabolic pathways such as methionine adenosyl transferase 2A (MAT2A) [127,128]. Downregulated miR-199a-5p targets and inhibits the polyamine metabolic enzyme spermidine/spermine N1-acetyltransferase (SAT-1) [129]. hsa-miR-1-3p 3’ UTR binds with ornithine decarboxylase (ODC), spermine synthase (SPMS) and SAT-1 [130].

## 5. Cardiometabolic Diseases and Their Regulation by MiRNA

Cardiometabolic diseases are multifactorial diseases that represent a combination of metabolic abnormalities characterized by dysregulated glucose tolerance, insulin resistance, hyperinsulinemia, type 2 diabetes, hypertension, dyslipidemia including high-serum triglyceride and low-serum high-density lipoprotein), atherosclerosis, and central adiposity. A wide variety of responsible factors, including changes in the environment, diet, and genetic and epigenetic factors, may contribute to the incidence of cardiometabolic complications. Most importantly, perturbations in the homeostatic metabolic regulation of fatty acids and glucose are correlated with the majority of cardiometabolic risk factors in type 2 diabetes mellitus (T2D), obesity, heart attack, stroke, and atherosclerosis [131,132]. The abnormal expression of miRNA has been associated with pathophysiologic events involved in the development and progression of chronic cardiometabolic diseases. This part of the review discusses the miRNA-mediated regulation of metabolism in prevalent cardiometabolic diseases or conditions. Table 1 describes the role of miRNAs and their respective direct target transcripts in the dysregulation of cardiac metabolism-mediated diseases, including lipid dyshomeostasis-mediated atherosclerosis, hyperlipidemia, T2DM, stroke, and obesity.

## 6. MiRNA Targeting Therapeutic Approaches

MicroRNAs, negative regulators of gene expression, have multiple target genes with homologous functions. Consequently, modifying a single miRNA can regulate the complete gene network and alter complex disease phenotypes, which is also considered a challenge for miRNA-based therapeutics, including the inhibition of specific miRNA that can target multiple genes and has multiple fundamental processes of cells. Various tools are available to specifically target miRNA pathways, such as modified antisense oligonucleotide (ASO), which targets the mature miRNA sequences; antimiRs, which mitigate the abnormally expressed miRNAs; miRNA mimics, which increase the miRNAs levels; miRNA sponges, which reduce miRNA levels; and erasers, which inhibit miRNA activity. Based on either sense or the anti-sense effects of miRNAs, therapeutic approaches will be selected for favorable gene expression, either gene activation or suppression. miRNA mimics could be used to decrease the expression of specific target genes; whereas, to upregulate the target gene expression, the miRNA inhibitors would be helpful [148,149,150]. The preclinical and clinical trials of miRNA therapeutic approaches are summarized in Table 2.

Increasing the production of high-density lipoprotein (HDL) is considered a potential therapeutic approach to treating most cardiometabolic diseases. Recent studies proposed that therapeutic targeting of miRNA offers an alternative strategy to increase HDL levels by upregulating ABCA1 expression. Locked nucleic acids (LNA) anti-miRNA antisense oligonucleotides are emerging as a therapeutic option for various cardiometabolic diseases [29,30,170]. Furthermore, targeting the exosomal miRNAs is the best therapeutic tool for managing diabetes-related cardiovascular complications [171,172]. Based on the available data, we recommend the evaluation of a more pronounced panel of multiple miRNAs for therapeutic intervention and the exploration of advances in targeting and the delivery efficacy of anti-miRNA antisense oligonucleotides, validation of the efficacy of various anti-miR chemistries, including tiny LNA, LNA-DNA mixmer, antagomir, 2’-F-2’-MOE mixmer for therapeutics and more clinical studies that are sufficient to modify existing clinical practices for the management of cardiometabolic diseases.

## 7. Limitations of this Review

We are aware that our review has two significant limitations. The first is delivering only a brief note on miRNAs’ mechanistic role in cardiometabolic diseases. The second is the lack of discussion on the diagnostic potential of the identified miRNAs. These limitations underline the unavailability of sufficient knowledge about the miRNA-specific biomarkers and interference of miRNAs in cardiometabolic diseases in the current literature.

## 8. Future Research Directions

It is now known that miRNAs have a potential role in cardiometabolic diseases and abnormalities by targeting and regulating many host genes and dysregulating the cardiac energy substrate metabolism. This review recommends that more research be undertaken to thoroughly understand the role of miRNAs and their inter-relationship with cardiometabolic disease targets, to identify the exact sources of circulating miRNAs, and to improve the studies regarding the assessment of the diagnostic and therapeutic potential of miRNAs in cardiometabolic diseases. Although miRNAs meet many of the needed criteria of an ideal biomarker, including high sensitivity, specificity, and accessibility, the utility of individual biomarkers for a cardiometabolic abnormality has comparatively low sensitivity. Therefore, future studies should focus on the effects of a combination of multiple miRNAs in the diagnosis of cardiometabolic diseases to enhance diagnostic and prognostic accuracy. Future investigations may identify a panel of miRNA biomarkers for every cardiac disease, and accordingly, the therapeutics may be available soon from future research. In addition, the expression profile of miRNAs in circulation may vary based on the different phases of cardiometabolic diseases. Therefore, future research should directly provide prognostic evidence of miRNA expression levels in cardiometabolic diseases. Large-scale prospective studies are needed to authenticate and further characterize the correlation of miRNAs in cardiac diseases.

Moreover, more clinical studies are highly recommended to illustrate and reveal the clinical cut-off, circadian fluctuations, and knowledge of the specimen-specific result dependency to confirm the efficacy of miRNAs as a biomarker. Future studies should provide guidance on standardizing the appropriate dosing regimen to overcome the off-target effects in inappropriate tissues and to validate the efficacy of various anti-miR chemistries in therapeutic development for cardiometabolic diseases. Finally, a group of miRNAs identified in the early and late stages of the disease will be presented, and those biomarkers will diagnose the early stage of cardiometabolic diseases and disease progression and serve as therapeutic targets.

## 9. Conclusions

Many studies have generated important clues about miRNAs and their essential roles in regulating cardiac metabolism. In this review, we systematically explained the impacts of miRNAs on the metabolism of cardiac energy-providing substrates and the development of cardiometabolic diseases. Therefore, miRNAs play essential roles in dysregulated cardiac metabolism-driven diseases, and miRNA-targeting therapeutics might be helpful to aid conventional therapeutics. The research articles reviewed here underscore the importance of understanding the complete miRNA biology in cardiac metabolism to identify specific biomarkers and broaden the treatment strategies for cardio-metabolic diseases. There is an immense amount of interest in initiating and progressing the miRNA targeting therapeutic strategies for treating cardiometabolic diseases, and the literature recommends that miRNA-associated combination therapy may aid the prognosis for patients with cardiac diseases. More mechanical studies and evaluation of miRNAs in cohort studies are needed to confirm and declare the clinical usage of miRNAs as diagnostic and prognostic biomarkers and therapeutic targets. This review adds advanced systematic knowledge to the literature, including the regulatory pattern of miRNAs in the metabolism of cardiac energy-providing substrates, and it proposes a highly pronounced panel of miRNAs specific for cardiometabolic abnormalities and recommends new perspectives on promising miRNA targeting therapeutic intervention.

## Figures and Tables

**Figure 1 ijms-24-00050-f001:**
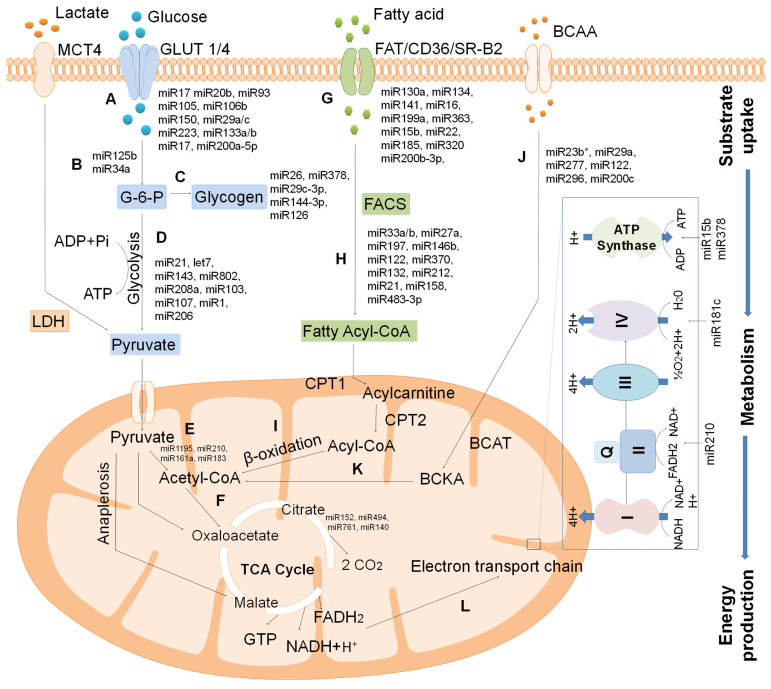
Role of miRNAs in cardiac energy substrate metabolism. (A) miRNA-regulated glucose transportation into cardiomyocytes by GLUT1 or GLUT4. (B) miR-125b and miR34a regulates the hexokinase activity. (C) Glucose 6- phosphate undergoes glycogen synthesis under the regulation of miR26, miR378, miR29c-3p, miR144-3p, and miR126. (D) Glucose metabolism by glycolysis under the regulation of miRNAs. (E) The resulting pyruvate is converted to acetyl-CoA, and (F) enters into the TCA cycle to produce reducing equivalents. (G) Free fatty acids enter into the cardiomyocytes by miRNAs and form the fatty acyl- CoA (H) to be involved in β-oxidation to form acetyl-CoA (I). (J) BCAA was transported into mitochondria, oxidizing to form acetyl-CoA (K). (L) After the TCA cycle, the reducing equivalents undergo oxidative phosphorylation to generate chemical energy, ATP. MCT4: Monocarboxylic anion transporter 4; G-6-P: Glucose -6-phosphate; LDH: lactate dehydrogenase; FAT: Fatty acid translocase, FAT: Fatty acid translocase, FACS: Fatty-acyl-CoA synthase; CPT1: carnitine palmitoyltransferase. BCAT: Branched-chain α ketoacids; BCAT: Branched-chain amino acid aminotransferase.

**Table 1 ijms-24-00050-t001:** Role of MiRNA in Cardiometabolic Disease/Complications by Dysregulated Cardiac Metabolism.

Cardiometabolic Complications/Diseases	miRNA/Host Gene	∆ Expression	Target Transcript	Mechanism of Dysregulated Cardiac Metabolism	Types of Model Studied and Pathological Features	Reference
Atherosclerosis (lipid dyshomeostasis- mediated)	miR-33a/ *SREBP2*, miR-33b/*SREBP1*	Upregulation	CROT, CPTA1α, HADHB	Downregulates fatty acid β-oxidation enzymes	HepG2 cells, THP1 cells, Y1 cells	[133]
			ABCA1	Downregulates the cholesterol transporter ABC transporter Decreases the plasma high-density lipoprotein (HDL) levels	Anti-miR33-treated mice/ Increases atherosclerotic plaques, inflammatory marker gene expression	[134]
			CROT, CPTA1α, HADHB, AMPKα, IRS2, SIRT6	Reduces fatty acid oxidation and insulin signaling	Huh7 cells/ Increased fatty acid accumulation	[135]
	miR-370, miR-122	Upregulation	CPTA1α, DGAT2	Downregulates CPTA1α expression Decreases the rate of β-oxidation Affects FAS and ACC-1 expression	HepG2 cells	[105]
	miR-122	Upregulation	CPTA1α	Downregulates the enzymes of lipid metabolism including HMGCS, HMGCR, and 7-dehydrocholesterol reductase (DHCR7)	Locked nucleic acid (LNA)- antimiR-122 treated African green monkey and mice	[136,137]
	miR-30	Downregulation	LPGAT1, MTP	Reduces lipid synthesis, triglyceride secretion	APOE knockout mice/Decreases atherosclerotic plaques	[107]
Hyperlipidemia	miR-148a/ *R3HDM1*	Upregulation	ABCA1	Downregulates the expression of nuclear receptors (PPARs, PGC-1α), LDLR, AMPK and ABCA1 essential for fatty acid metabolism by SREBP1 mediated pathway Increases circulating level of LDL-C	miR-148a-deficient mice, APOE- knockout mice	[138]
	miR-128	Upregulation	LDLR	Represses the sirtuin 1 (SIRT1) gene expression Increases circulating level of LDL-C	miR-128-deficient mice	[139]
	miR-30	Downregulation	LPGAT1, MTP	Inhibits microsomal transfer protein (MTP) Decreases lipid synthesis and lipoprotein secretion Decreases serum level of triglycerides and LDL-C	Anti-miR-30c treated mice	[107]
	miR-140-5p	Upregulation	LDLR	Deceases LDLR expression Reduces LDL-C uptake	Anti-miR-104-5p (Simvastatin) transfected HepG2 cells	[140]
	miR-27a miR-27b	Upregulation	ABCA1 PPARγ	Reduces lipid metabolism by downregulating the lipid metabolic genes (Fatty acid synthase, SREBP-1, SREBP-2, PPARα, PPARγ) and ApoA1, ApoB100 and ApoE3	293T cells	[103]
	miR-378/*PGC-1β*	Upregulation	CRAT, MED13, ERRγ, GABPA, IGF1R, ABCG1	Counterbalances the metabolic functions of PGC-1β	miR-378 knockout mice	[110]
Dyslipidemia	miR-27b	Upregulation	GPAM, ANGPTL3	Downregulates lipid metabolism by repressing lipid metabolic genes such as PPARγ, GPAM, and ANGPTL3 Increases levels of plasma lipid	APOE knockout mice	[141]
Type II diabetes mellitus and obesity	miR-29	Upregulation	FOXA2	Inhibits the activation of lipid metabolism genes including PPARGC1A, HMGCS2, and ABHD5 Inhibits the lipid metabolism	Zucker diabetic fatty (*fa*/*fa*) rats	[142]
	miR-26a	Downregulation	Acc1, Acc2, Acly, Dgat2, Fasn, Lipc, Srebf1	Improves insulin sensitivity Decreases glucose production and fatty acid synthesis Protects from obesity-induced metabolic complications	miR-26a transgenic obese mice	[143]
	miR-375	Upregulation	Insulin, Mtpn	Downregulates glucose-stimulated insulin secretion Downregulates phosphoinositide-dependent protein kinase-1 Reduces the activation of AKT and GSK3	miR-375 knockout mice, *ob*/*ob* mice (obese, insulin resistant, T2DM)	[144]
	Let-7	Upregulation	INSR, IRS2	Represses the gene expression of INSR, IRS2 Impairs global glucose homeostasis Blocks insulin signaling	Diet-induced obese mice	[145]
	miR-33/*SREBP2*	Upregulation	ABCA1	Translational repression of ABCA1 to reduce cellular cholesterol export Downregulates fatty acid β-oxidation enzymes	Drosophila melanogaster	[133]
Non-alcoholic fatty liver disease (NAFLD)	miR-21	Upregulation	HMGCR	Downregulates the lipid and triglyceride metabolism Increases serum level of mRNA and protein HMGCR	Palmitic acid and oleic acid treated HepG2	[146]
	miR-122	Upregulation	FASN, ACC, SCD1, SREBP	Enhances the serum levels of lipid and triglyceride Decreases fatty acid oxidation	miR-122 knockout mice	[147]

FOXA2: Forkhead box protein A2; LPGAT: Lysophosphatidylglycerol Acyltransferase; R3HDM1: R3H Domain Containing 1; LDLR: low-density lipoprotein receptor; APOE: apolipoprotein E; LDL-C: Low-density lipoprotein cholesterol; MED13: Mediator Complex Subunit 13; GABPA: GA Binding Protein Transcription Factor Subunit Alpha; GPAM: Glycerol-3-Phosphate Acyltransferase—Mitochondrial; ANGPTL3: Angiopoietin-like protein 3; MTP: Microsomal triglyceride transfer protein; FOXA2: Forkhead box protein A2; PPARGC1A: Peroxisome proliferator-activated receptor gamma coactivator 1-alpha; ABHD5: Abhydrolase Domain Containing 5; SCD1: Stearoyl-CoA desaturase 1.

**Table 2 ijms-24-00050-t002:** miRNA with Therapeutic Potential in the Development Stage.

Therapeutic Drug	Target MiRNA	Investigating Diseases	Stage of Trials	References
Miravirsen (SPC3649)	miR-122	Hepatitis C virus infection	Phase II clinical trials	[151]
MGN-1374	miR-15 and miR-195	Post-myocardial infraction	Preclinical stage	[152]
AZD4076	miR-103/107-3p	Type II diabetes with non-alcoholic fatty liver disease Type II diabetes with non-alcoholic steatohepatitis	Phase I/IIa	[153]
MRX34	miR-34a	Different types of cancer	Phase I clinical trials	[154]
MGN-6114	miR-92	Peripheral arterial disease	Preclinical stage	[155]
MesomiR-1	miR-16	Lung cancer, Mesothelioma	Phase I clinical trial	[156]
Remlarsan (MRG-201)	miR-29	Fibrosis	Phase I clinical trial	[157]
RGLS4326	miR-17-5p	Polycystic kidney disease (PKD)	Phase I clinical trials	[158]
CDR132L	miR-132-3p	Stable heart failure	Phase I clinical trial	[159]
MGN-4220	miR-29	Cardiac fibrosis	Preclinical stage	[160]
MGN-5804	miR-378	Cardiometabolic disease	Preclinical stage	[161]
RG-101	miR-122	Virus infection	Phase IB clinical trials	[162]
MRG-107	miR-155	Amyotrophic lateral sclerosis (ALS)	Entering in clinical trial	[163]
MRG-110	miR-92a	Ischemia	Phase I clinical trial	[164]
MGN-9103	miR-208	Chronic heart failure	Preclinical stage	[165]
Cobomarsen (MRG-106)	miR-155	Cutaneous T-cell lymphoma (CTCL)	Phase I clinical trials	[166]
MGN-2677	miR-143/145	Vascular disease	Preclinical stage	[167]
TargomiR	miR16-5p	Malignant pleural mesothelioma	Phase I clinical trial	[168]
RG-012 Lademirsen	miR-21-5p	Nephropathy	Preclinical stage	[169]

## Data Availability

Not applicable.

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
