# Peer review of "Cardiac Metabolism and MiRNA Interference"

_ijms, 2022, doi:10.3390/ijms24010050_

Round 1

Reviewer 1 Report

This review is categorized into four platforms, which encompass i) a review of the fundamental 27 mechanism of cardiac metabolism, ii) a divulgence of the regulatory role of specific miRNAs on 28 cardiac metabolic pathways, iii) an understanding of the association between miRNA and impaired 29 cardiac metabolism, and iv) summary of available miRNA targeting therapeutic approaches.

Include in article

0 -   Describe the methodology used to make this review. Include the methodology used to do the review: database, years of search, keywords.

1 - Compare and contrast your study with others in the most relevant world literature, particularly the recent literature.

2 - What new information is sufficient to modify existing clinical practice?

3 - What are the conclusions and implications for current practice, and particularly for future research that may have a significant impact on clinical decisions?

4 - What does this study add to the literature?

5 - At the end of the Discussion, under the subheading "Limitations," review the limitations of your study.

6 - At the end of the limitations, under the subheading " Future directions".

7 - Conclusion

Take special care to draw your conclusions only from your results and verify that your conclusions are firmly supported by your data

8– References

Update 

9 - Include: Author Contributions and Conflict of Interest

Author Response

We thank the reviewer for their valuable time and constructive comments.

Describe the methodology used to make this review. Include the methodology used to do the review: database, years of search, keywords.

Response: We thank the reviewer for this constructive comment. We have included the detailed methodology under section 2 of the revised manuscript

Compare and contrast your study with others in the most relevant world literature, particularly the recent literature.

Response: As per the comment, we have included the statement comparing and contrasting our study with recent literature in section 1 of the revised manuscript.

What new information is sufficient to modify existing clinical practice?

Response: As suggested by the reviewer, we included the novel information expected to modify the existing clinical practice in section 6 of the revised manuscript.

What are the conclusions and implications for current practice, and particularly for future research that may have a significant impact on clinical decisions?

Response: As suggested, we provided a detailed description regarding the implications of the current practice to direct future research in sections 6 and 8 of the revised manuscript.

What does this study add to the literature?

Response: This review adds advanced systematic knowledge to the literature, including the regulatory pattern of miRNAs in the metabolism of cardiac energy-providing substrates, propose the highly pronounced panel of miRNAs specific for cardiometabolic abnormalities, and recommends new perspectives on promising miRNA targeted therapeutic intervention. We have discussed it in detail in the conclusion section of the revised manuscript.

At the end of the Discussion, under the subheading “Limitations”, review the limitations of your study.

Response: We have now provided a section in the revised manuscript that describes the limitation of our review.

At the end of the limitations, under the subheading “Future directions”.

Response: We have now included a section in the revised manuscript to describe the future directions of miRNA-based therapy and prognosis for cardiometabolic diseases.

Conclusion: Take special care to draw your conclusions only from your results and verify that your conclusions are firmly supported by your data.

Response: As suggested, we updated our conclusions based on our knowledge from the available literature.

References: Update                                          

Response: We provided appropriate updated references as needed.

Include: Author Contributions and Conflict of Interest

Response: We have included author contributions and conflict of interest in the revised manuscript.

Reviewer 2 Report

The review manuscript by Sumaiya et al., overview a very relevant, interesting, and novel topic, namely the role of miRNA interference. The article is well and comprehensively written. I suggest the paragraph 2 describing cardiac metabolism to be shorten for better readability. Overall, the manuscript is very detailed, includes very recent findings in the field, and after a minor revision, I suggest being consider for publication. 

Author Response

We thank the reviewer for their valuable time and positive comment.

Response: As suggested, we have shortened paragraph 2 in the revised manuscript.

Round 2

Reviewer 1 Report

The authors responded adequately to the comments and there was a significant improvement in the manuscript.